# Perirenal Fat Thickness Significantly Associated with Prognosis of Metastatic Renal Cell Cancer Patients Receiving Anti-VEGF Therapy

**DOI:** 10.3390/nu14163388

**Published:** 2022-08-18

**Authors:** Kang Ning, Zhen Li, Huiming Liu, Xi Tian, Jun Wang, Yi Wu, Longbin Xiong, Xiangpeng Zou, Yulu Peng, Zhaohui Zhou, Fangjian Zhou, Chunping Yu, Junhang Luo, Hailiang Zhang, Pei Dong, Zhiling Zhang

**Affiliations:** 1Department of Urology, Sun Yat-sen University Cancer Center, Guangzhou 510080, China; 2State Key Laboratory of Oncology in Southern China, Collaborative Innovation Center for Cancer Medicine, Guangzhou 510060, China; 3Department of Medical Imaging, Sun Yat-sen University Cancer Center, Guangzhou 510080, China; 4Department of Urology, Fudan University Shanghai Cancer Center, Shanghai 201102, China; 5State Key Laboratory of Ophthalmology, Zhongshan Ophthalmic Center, Sun Yat-sen University, Guangzhou 510080, China; 6Department of Urology, The First Affiliated Hospital, Sun Yat-sen University, Guangzhou 510080, China

**Keywords:** body composition, adipose tissue, renal cell carcinoma, prognosis, anti-VEGF therapy

## Abstract

Although high body mass index (BMI) was reported to associate with a better prognosis for metastatic renal cell cancer (mRCC) patients receiving anti-vascular endothelial growth factor (anti-VEGF) therapy, it is an imperfect proxy for the body composition, especially in Asian patients with a lower BMI. The role of visceral adipose tissue (VAT), subcutaneous adipose tissue (SAT), and perirenal fat thickness (PRFT) in mRCC patients was still unknown. Therefore, a multicenter retrospective study of 358 Chinese mRCC patients receiving anti-VEGF therapy was conducted and their body composition was measured via computed tomography. We parameterized VAT, SAT and PRFT according to their median value and BMI according to Chinese criteria (overweight: BMI ≥ 24). We found VAT, SAT, and PRFT (all *p* < 0.05) but not BMI, significantly associated with overall survival (OS) and progression-free survival (PFS). Multivariate Cox analysis identified PRFT was the independent predictor of OS and PFS, and IMDC expanded with PRFT showed the highest C-index in predicting OS (OS:0.71) compared with VAT, SAT, and BMI. PRFT could increase the area under the curve of the traditional International Metastatic Renal Cell Carcinoma Database Consortium (IMDC) model in OS (70.54% increase to 74.71%) and PFS (72.22% increase to 75.03%). PRFT was introduced to improve the IMDC model and PRFT-modified IMDC demonstrated higher AIC in predicting OS and PFS compared with the traditional IMDC model. Gene sequencing analysis (*n* = 6) revealed that patients with high PRFT had increased angiogenesis gene signatures (NES = 1.46, *p* = 0.04) which might explain why better drug response to anti-VEGF therapy in mRCC patients with high PRFT. The main limitation is retrospective design. This study suggests body composition, especially PRFT, is significantly associated with prognosis in Chinese mRCC patients receiving anti-VEGF therapy. PRFT-modified IMDC model proposed in this study has better clinical predictive value.

## 1. Introduction

Significant advancements in therapies for patients with metastatic renal-cell-carcinoma (mRCC) have been made over the past few years [1,2]. Antagonists of vascular endothelial growth factor (anti-VEGF) agents including sunitinib, axitinib are fundamental components of first-line treatment for mRCC [3]. The International Metastatic renal-cell-carcinoma Database Consortium (IMDC) risk model was proposed to guide patient selection for clinical trials, counseling, and risk-specific treatment [4]. Although the value of the IMDC model has been confirmed in mRCC patient management [5], not all patients could benefit from anti-VEGF therapies, and some may even experience severe adverse events [6]. Therefore, many new predictors associated with a patient’s prognosis are still being explored.

The body mass index (BMI), a traditional indicator of general obesity, was reported to be associated with better overall survival (OS) and progression-free survival (PFS) in mRCC patients receiving anti-VEGF therapies [7]. However, most studies between BMI and mRCC have been conducted among Caucasians, who tend to have a higher BMI than Asians. In Asian patients who tend to have lower BMI, the performance of BMI to identify excess adiposity has a high specificity and low sensitivity, suggesting an under-detection of obesity [8]. Besides, there was high variability in fat mass and muscle mass within all strata of BMI in cancer patients [9], and BMI failed to reflect more specific measures of body composition [10]. For example, a Japanese study has reported that Asian patients with lower BMI had a better prognosis than those who were overweight in gastrointestinal cancer surgery [11], and this finding was quite different from previous American studies of gastrointestinal cancer among Caucasians [12].

Computed tomography (CT)-based body composition measures can distinguish between visceral adipose tissue (VAT), subcutaneous adipose tissue (SAT), and perirenal fat thickness (PRFT) [13]. Since BMI has been shown to be an imprecise measurement of fat-free and fat mass, body cell mass, and fluids, CT-based body composition measure was a better way to evaluate the fat distribution and nutrient levels [14]. Besides, fat in different areas of the body also has different biological properties. VAT and SAT have different origins and the developmental heterogeneity may determine the functional heterogeneity [15]. Compared with SAT, VAT has more blood vessels, innervation, and inflammatory cells, is metabolically more active, more sensitive to lipolysis, and more insulin-resistant [16]. Although perirenal fat is a type of VAT adjacent to the kidneys, it is reported to originate from both adipocyte precursor cells and mature adipocytes; different from the origins of other components of VAT [17,18]. Chen et al. found that the anatomical position and developmental heterogeneity of perirenal fat make it more sensitive to predict the development of chronic kidney disease than VAT and SAT in patients with diabetes [19]. However, the association between fat in different areas of the body and anti-VEGF therapy in patients with mRCC remains unclear. Given the superiority of CT-based body composition and the special biological properties of fat, it is worth exploring whether fat in different areas of the body is a more valuable predictor in mRCC compared with BMI.

In this study, a multicenter retrospective cohort was conducted to analyze the impact of BMI and body composition on the OS and PFS in Chinese mRCC patients receiving anti-VEGF therapies. The prognostic value of fat in different areas of the body was evaluated and compared. The role of body composition played in the IMDC model was also identified in this study. Although limitations such as selection bias and recalling bias were inevitable in retrospective studies, our results may provide significance for the screening of predictors associated with patients’ prognosis in mRCC.

## 2. Materials and Methods

### 2.1. Clinical Data Collection

The data of 358 mRCC patients receiving first-line anti-VEGF therapy from May 2008 to September 2020 were retrospectively reviewed in three large-volume hospitals. Patients without a CT scan for body composition evaluation or less than a one-year follow-up were excluded. The pretreatment data on BMI, OS, PFS, and CT scans of the 358 patients were retrieved. PFS was defined as the time from the start of anti-VEGF therapy to the date of disease progression on radiological imaging or the date of death as a result of any cause or was censored at the date of last imaging. OS was defined as the time from the start of anti-VEGF therapy to the date of death as a result of any cause or was censored at the date of the last follow-up. Risk factors in the IMDC model included TDT of less than one-year, Karnofsky performance status of less than 80% (A measure of self-care in cancer patients, and less than 80% means the patients have been unable to live and work normally), neutrophil greater than the upper limit of normal (ULN), platelets greater than ULN, corrected serum calcium greater than the ULN and serum hemoglobin less than the lower limit of normal (LLN). According to the number of risk factors, all patients were classified into three risk groups by the IMDC risk model: favorable (0 risk factor), intermediate (1–2 risk factors), and poor (>2 risk factors) [4].

### 2.2. Measures of CT-Based Body Composition

According to previously established methods, PRFT was evaluated at the renal vein level to reflect the level of adipose tissue around the kidneys [20]. PRFT was defined as the sum of lateral and posterior perirenal fat thicknesses (Appendix A). Skeletal muscle (SM), VAT, and SAT were measured by analyzing single-slice abdominal CT scans at L3 using the medical imaging software SliceOMatic version 5.0 (TomoVision, Montreal, QC, Canada). SM, VAT, and SAT at L3 were highly correlated with total body skeletal muscle and total body adipose tissue volume [21]. CT-based body composition was identified based on the Hounsfield Units via region growing image segmentation algorithm (Appendix A). Skeletal muscle index (SMI) was calculated by dividing SM (cm^2^) by height^2^ (m^2^). Total adipose tissue (TAT) equaled VAT plus SAT. All CTs used for measures of body composition were performed within 3 months prior to receiving anti-VEGF therapy.

### 2.3. NanoString Digital Spatial Profiler

The NanoString DSP technology was performed with the help of Fynn Biotechnologies Ltd. (Jinan, China). Briefly, the FFPE slides from 6 patients were deparaffined, subjected to antigen retrieval procedures, and incubated in 1 µg mL^−1^ proteinase K (Thermo Fisher Scientific, AM2546, Waltham, MA, USA) in PBS for 15 min at 37 °C. The slides were then incubated with GeoMx RNA detection probes in Buffer R overnight at 37 °C. Sequencing libraries and PCR reactions were performed according to the manufacturer’s instructions. PCR reactions were pooled and purified twice using AMPure XP beads (Beckman Coulter, A63881, Brea, CA, USA). Pooled libraries were sequenced on NextSeq 2000 platform.

### 2.4. Quantitative Reverse Transcription Polymerase Chain Reaction (RT-qPCR)

Perirenal fat was collected from fifty-six patients for RT-qPCR to test the gene expression of leptin, interleukin-6 (IL-6), tumor necrosis factor (TNF), and adiponectin, which were reported to play an important role in the obesity paradox [22]. All samples of perirenal fat were from fifty-six patients with RCC who underwent radical nephrectomy at our hospital between July and August 2021 (*n* = 61). RNA extraction was performed as described above. Total RNA (1 mug) was used as a template to synthesize complementary DNA (cDNA) using a PrimeScript RT Reagent Kit with cDNA Eraser (Takara Biotechnology, Kusatsu, Japan). Subsequently, qRT-PCR was performed using SYBR Premix Ex Taq (Takara Bio Inc., Kusatsu, Japan). All qRT-PCR assays were performed using the ABI 7900 system (Applied Biosystems, Waltham, MA, USA). Only RNA of qualified quality was included in the analysis (*n* = 56).

### 2.5. Statistical Analysis

Statistical analyses were performed using the R software (Version 4.0.3, Auckland, New Zealand) and *p* < 0.05 were considered statistically significant for the statistical analyses. Statistically significant genes in sequencing were obtained by an adjusted *p* value threshold of <0.05 and |log2 (fold change)| > 1. Wilcoxon signed-rank test was used to compare the differences in gene expression. Univariate Cox proportional hazards regression analysis was performed to evaluate the predictive value of all potential predictors in OS and PFS. Continuous variables were transformed into dichotomous variables with the median as the cut-off value. In multivariable analysis, all the parameters related to CT-based body composition were adjusted for gender and IMDC score. The sensitivity and specificity of the model were assessed by the receiver operating characteristic (ROC) curve (3 years). The improved model fit was measured by the concordance index (C-index), area under curve (AUC), and Akaike Information Criterion (AIC). Conclusively, the higher C-index and AUC or the lower the AIC, the better the model fit.

## 3. Results

### 3.1. Patient Characteristics and Outcomes

A total of 358 mRCC patients were eligible for this analysis, of whom 183 (51.1%), 68 (19.0%), 60 (16.8%), 30 (8.4%), and 17 (4.7%) patients were treated with sunitinib, axitinib, pazopanib, sorafenib, and bevacizumab, respectively (Table 1). 249 (69.9%) patients were diagnosed with clear cell RCC, and 289 (80.7%) patients underwent nephrectomy. The lungs (47.5%) were the most frequent metastatic site. Until the last follow-up, 146 (40.8%) patients had received immunotherapy. The median of BMI, PRFT, VAT, and SAT was 23.0 (interquartile range, IQR: 21.0–24.9) kg/m², 1.6 (IQR: 1.1–2.6) cm, 81.9 (IQR: 39.3–138.4) cm², and 100.2 (IQR: 66.1–145.4) cm², respectively. At the point of final analysis, 117 (32.6%) patients had died, and 267 (74.6%) patients had tumor progression since the initiation of anti-VEGF therapy. The median OS and PFS for the entire cohort of 358 patients were 22.0 (IQR: 13.0–37.0) months and 9.1 (IQR: 4.8–16.1) months.

### 3.2. Univariable Analysis

Correlation analysis showed that several parameters of CT-based body composition were correlated (Appendix A). In univariable analysis (Table 2), we found that the Karnofsky score < 80 (HR = 2.86, 95%CI: 1.96–4.18, and HR = 1.92, 95%CI: 1.47–2.52, respectively), TDT < 1 year (HR = 2.46, 95%CI: 1.56–3.87), and HR = 1.30, 95%CI: 1.00–1.69), respectively), lower hemoglobin (HR = 1.82, 95%CI: 1.25–2.66), and HR = 1.58, 95%CI: 1.23–2.04), respectively), and higher corrected calcium (HR = 4.01, 95%CI: 2.39–6.74), and HR = 2.41, 95%CI: 1.58–3.69), respectively) significantly predicted worse OS and PFS, respectively. However, neutrophils and platelets as the factors in the IMDC risk model were not associated with OS and PFS. Further, we observed that immunotherapy was a protective factor strongly associated with better OS (HR = 0.47, 95%CI: 0.32–0.70) but not PFS (HR = 0.93, 95%CI: 0.73–1.19).

Of the parameters related to body composition (Table 2), PRFT (Figure 1B,D, HR = 0.53, 95%CI: 0.37–0.77, and HR = 0.75, 95%CI: 0.59–0.95, respectively), VAT (Appendix A, HR = 0.60, 95%CI: 0.42–0.87, and HR = 0.74, 95%CI: 0.58–0.94, respectively), SAT (Appendix A, HR = 0.47, 95%CI: 0.33–0.69, and HR = 0.69, 95%CI: 0.54–0.88, respectively), and TAT (Appendix A, HR = 0.56, 95%CI: 0.39–0.82, and HR = 0.70, 95%CI: 0.55–0.89, respectively) were identified as significantly associated with OS and PFS in patients receiving anti-VEGF therapy. High BMI (≥24, overweight standards for Chinese population [23]) was not a significant predictor for OS (*p* = 0.46) and PFS (*p* = 0.38) (Figure 1A).

### 3.3. Multivariable Analysis

In multivariable Analysis, PRFT was proved to be the only independent predictor of OS (HR = 0.57, 95%CI: 0.35–0.93) and PFS (HR = 0.78, 95%CI = 0.61–0.98) in all body composition (Appendix A). IMDC model is a classical model of mRCC, and we tried to incorporate each body composition parameter into the model, and explore which parameter could improve the IMDC model best in multivariate cox regression analysis (Appendix A). The C-index for the IMDC risk model significantly demonstrated OS (0.69, 95% confidence interval, 95%CI: 0.63–0.74) and PFS (0.61, 95%CI: 0.57–0.65). When PRFT was incorporated into the IMDC risk model, the C-index increased to 0.71 for OS and 0.62 for PFS. Other prognosticators, such as SM, VAT, SAT, and TAT could not improve the C-index of the IMDC model. However, BMI seemed to increase the C-index of IMDC for predicting PFS (0.62). In ROC curve analysis of 3-year OS and PFS (Figure 2), the AUC of the traditional IMDC model was 70.54% and 72.22%. PRFT-modified IMDC model had an improved AUC for predicting OS (74.71%) and PFS (75.03%). However, the BMI-modified IMDC model didn’t significantly improve the prediction efficiency of OS (71.96%) and PFS (70.45%).

### 3.4. PRFT-Modified IMDC Model

All patients were classified into three risk groups by IMDC risk group: favorable group (*n* = 60), intermediate group (*n* = 192), and poor group (*n* = 106), and we further analyzed the association between PRFT and prognosis in different IMDC risk groups. Although PRFT was not associated with OS (Appendix A, HR = 0.75, *p* = 0.65) or PFS (Appendix A, HR = 1.22, *p* = 0.53) in the IMDC favorable group, it was a significant predictor for OS and PFS in the IMDC intermediate group (Appendix A, HR = 0.55, *p* = 0.02, and HR = 0.73, *p* = 0.049, respectively) and poor group (Appendix A, HR = 0.47, *p* = 0.02, and HR = 0.62, *p* = 0.04, respectively). Consequently, we proposed the PRFT-modified IMDC model, based from which the patients could be categorized into four risk groups: (1) Favourable: 0 risk factor (*n* = 60); (2) Intermediate-1: 1–2 risk factors and PRFT > median (*n* = 109); (3) Intermediate-2: 1–2 risk factors but PRFT ≤ median; more than 2 risk factors but PRFT > median (*n* = 122); (4) Poor: more than 2 risk factors and PRFT ≤ median (*n* = 67). PRFT modified IMDC model showed a higher degree of fit compared with the traditional IMDC model in predicting OS (Figure 3A,B, AIC: 1167.32 vs. 1181.53) and PFS (Figure 3C,D, 2640.79 vs. 2657.92).

### 3.5. Gene Sequencing Analysis in Different PRFT

To better explain the association between prognosis and PRFT, cancer-related gene sequencing from 6 patients with different PRFT was performed. The heat maps (Figure 4A) and a volcano plot (Figure 4B) of differential genes suggested that the expression of genes related to angiogenesis upregulate such as *JAG1*, *VEGFA*, and *TGFB1*. GSEA analysis identified that the RCC of patients with high PRFT had increased angiogenesis signatures (NES = 1.46, *p* = 0.04), epithelial-mesenchymal transition (EMT) signatures (NES = 1.53, *p* = 0.002), and transforming growth factor-β (TGF-β) signatures (NES = 1.42, *p* = 0.04) (Figure 4C).

It was reported that adipocyte cells secreted leptin, TNF, IL-6, and adiponectin to induce a change in tumor gene expression related to angiogenesis signatures [22]. For further investigations, perirenal fat tissues were collected from 56 patients for RT-qPCR. We found that the *LEP* (*p* = 0.005), *IL-6* (*p* = 0.03) and *TNF* (*p* = 0.02) were highly expressed in patients with high PRFT, while low expression of adiponectin (*ADIPOQ*) gene (*p* = 0.048) was observed (Figure 5A). Based on such findings, we hypothesized that the increased secretion of leptin, TNF, and IL-6 together with the decreased secretion of adiponctin in perirenal adipocytes could be associated with the upgrade of angiogenesis signatures in RCC and might enhance drug delivery and improve patients’ response to anti-VEGF therapy (Figure 5B).

## 4. Discussion

Better survival outcomes were found in patients with higher BMI who underwent targeted therapy [7,24], but those studies were almost conducted among Caucasians, who tend to have a higher BMI than Asians. Considering Asian populations generally have a lower BMI [10], the association between obesity and anti-VEGF therapy in RCC should be discussed further. CT-based body composition can distinguish adipose tissue distribution, as well as the quantity and quality of muscle, and can help refine our understanding of the association between fat in different parts of the body and cancer survival [13]. In this study, we found BMI was not significantly associated with OS and PFS in Asian mRCC patients, while PRFT as a new predictor has been found to predict the OS and PFS in mRCC patients receiving anti-VEGF therapy. The PRFT-modified IMDC model proposed in this study has better clinical predictive value. The underlying mechanism was speculated by sequencing and RT-qPCR, and the tumor from patients with high PRFT demonstrated increased angiogenesis signatures.

Although obesity and high-level fat were considered to be protective factors during anti-VEGF therapy in mRCC patients, obesity was also identified as a risk factor for the onset of RCC in several studies [25,26]. This interesting phenomenon was known as the “obesity paradox”, which has been also observed in other cancers including diffuse large B-cell lymphoma [27], and melanoma [28]. All these studies about the obesity paradox defined obesity by high-level BMI, however, the performance of BMI to identify obesity had a high specificity but low sensitivity, which could attenuate an obesity-disease association [29]. Besides, BMI is related to region and ethnicity. For example, it was shown that 60% of mRCC patients were overweight (defined as ≥25 kg/m², WHO criteria) in the IMDC cohort (study among the Caucasian population) [7], while only 38% of the patients were overweight (defined as ≥24 kg/m², Chinese criteria) in our study. Hence, BMI may not be a good way to explain the obesity paradox, while CT-based body composition was a good choice to better reflect the role of different parts of fat from onset to the treatment of RCC.

Many studies have reported that CT-based body composition was associated with survival in cancer patients [30,31]. However, there was some controversy in previous studies in mRCC. A meta-analysis identified that low SMI was associated with an increased risk of overall mortality in mRCC while VAT and perirenal fat were not consistently associated with patients’ survival [32]. High VAT was an independent prognostic factor for survival in some studies [33,34,35], while two studies have failed to yield significant results [36,37]. However, in the two articles with negative results, the sample size was limited, and not all mRCC patients underwent anti-VEGF treatment (some received cytokines [36] and some placebo [37]). Through analysis of our Chinese mRCC cohort which had a larger sample size than previous studies, we found that CT-based body composition, including VAT, SAT, SM, and perirenal fat, was significantly associated with OS and PFS (all *p* < 0.001). Due to the retrospective nature of this present study, prospective studies are needed to validate our findings.

We also found that PRFT, as a simple and reliable estimate of perirenal fat volume [38], could significantly improve the IMDC risk model compared with other CT-based body compositions. Especially in the intermediate and poor risk group of IMDC, the patient’s risk group could be changed when perirenal fat was included and patients with high PRFT were found to have better OS and PFS. Perirenal fat has origins in both adipocyte precursor cells and mature adipocytes different from the origins of other components of visceral fat and that developmental heterogeneity may determine the functional heterogeneity [17,39]. Perirenal fat adjacent to the kidneys was active in metabolism and adipokine secretion, which may play a special role in kidney diseases compared to fat in other regions [19]. It was reported that perirenal fat had a higher predictive value for chronic kidney disease than total, subcutaneous, or visceral fat in diabetes [19]. An animal experiment showed that perirenal fat directly causes renal artery endothelial dysfunction, which was partly mediated by tumor necrosis factor-α [40]. So, it would be interesting to further explore the gene expression difference of perirenal fat between PRFT low and high patients, which may help to explain the difference in prognosis.

Epidemiological investigation showed that obesity was significantly associated with RCC [26]. Many plausible biological mechanisms such as insulin resistance, insulin-like growth factors, and sex hormones pathways have explained how obesity leads to RCC [41,42]. However, our sequencing result reveals that patients with high PRFT had increased angiogenesis gene signatures which might enhance drug delivery and improve the patient’s response to anti-VEGF therapy in RCC. Sanchez et al. also found that VAT created regions of hypoxia which promotes angiogenesis and alterations of the tumor microenvironment that regulate RCC growth [39]. The secretion of adipokines (leptin and adiponectin) and cytokines (C-reactive protein, TNF, IL-6, IL-8) driven by adipose tissue in obesity can alter the expression and secretion of factors associated with angiogenesis in the primary tumor [22]. Due to the proximity to the kidneys, adipokines and cytokines from perirenal fat could not only influence insulin sensitivity and glucose and lipid metabolism but also directly affect renal hemodynamics and renal function [19]. We also found the high expression of genes of leptin, IL-6, and TNF and low expression of the adiponectin gene in patients with high PRFT, but it remains unclear whether the expression level and biological function of adipokines and cytokines from perirenal fat were different from those of other fat depots.

The present study had several limitations. First, because of the retrospective design, the time between pretreatment CT scan and anti-VEGF therapies was not standardized, but we believe that this might have not significantly impacted our study results because this time difference might have been too short to change the patient’s body composition. Second, the differences in pretreatment and additional therapies were described. For example, some patients received a combination of targeted and immunotherapy, which may have an impact on patients’ OS. Third, although we hope to put forward reasonable hypotheses of the clinical findings, the sequencing sample size (*n* = 6) was small and limited. Hence, the mechanism by which perirenal adipocytes secreting adipokines modifies the tumor microenvironment has not been well described and further studies are needed for clarification. Considering that this is the largest multicenter retrospective cohort study on the association between body composition and outcomes of anti-VEGF therapies in Asia, these findings could be used as references for guiding future clinical practices and scientific studies.

## 5. Conclusions

CT-based body composition, especially PRFT, was significantly associated with OS and PFS in mRCC patients receiving anti-VEGF therapy. Compared with VAT, SAT, and BMI, PRFT was an independent prognosticator for OS and PFS in Chinese mRCC patients. The predictive performance of the IMDC model was significantly improved when PRFT was incorporated into this risk model. The increase in angiogenesis signatures in patients with high PRFT may explain the increased susceptibility to anti-VEGF therapy, although the mechanism of the obesity paradox needs to be further studied.

## Figures and Tables

**Figure 1 nutrients-14-03388-f001:**
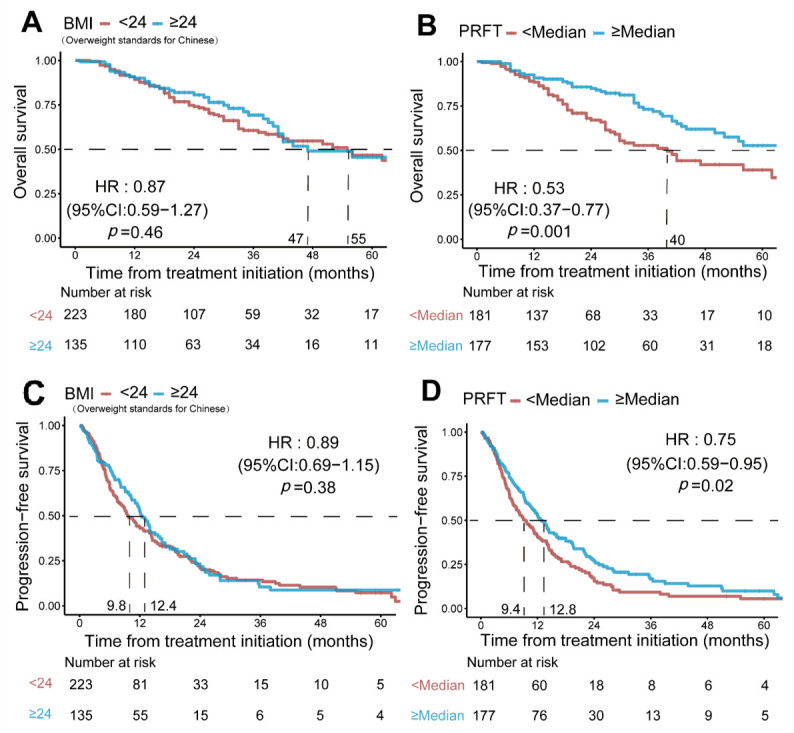
**Log-rank survival analysis of different BMI and PRFT in 358 patients treated with target therapy.** BMI was not a significant predictor in either OS (**A**) or PFS (**D**). While patients can be well grouped into different prognostic risk groups according to different PRFT (**B**,**C**). BMI: body mass index; PRFT: perirenal fat thickness; OS: overall survival. PFS: progression-free survival.

**Figure 2 nutrients-14-03388-f002:**
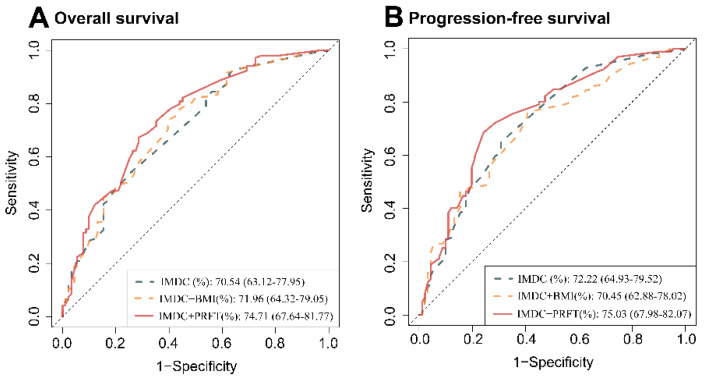
**The ROC analysis of OS and PFS in 358 patients treated with target therapy.** PRFT can improve the AUC of IMDC risk model for predicting of OS (**A**) and PFS (**B**) in patients with metastatic renal cell carcinoma. AUC: area under curve; BMI: body mass index; IMDC: International Metastatic Renal Cell Carcinoma Database Consortium. PRFT: perirenal fat thickness; OS: overall survival; PFS: progression-free survival; ROC: receiver operating characteristic curve.

**Figure 3 nutrients-14-03388-f003:**
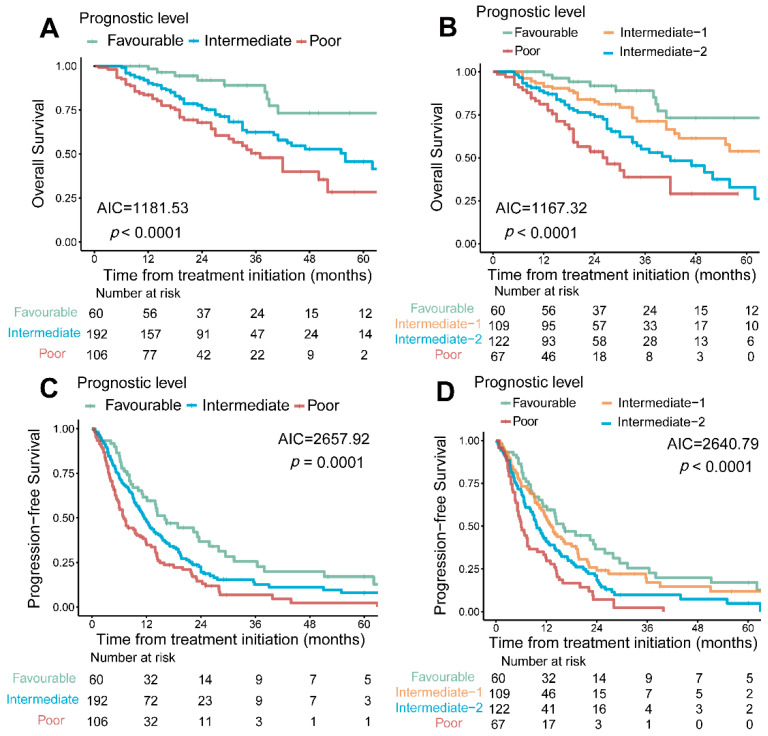
**PRFT-modified IMDC risk model compared with traditional IMDC risk model.** Traditional IMDC model including six risk factors: TDT, Karnofsky performance status, neutrophil, platelets, corrected serum calcium, hemoglobin. Combined with PRFT, we improved the traditional IMDC model. PRFT-modified IMDC risk model divided patients into four risk groups: favorable (0 risk factor), intermediate-1 (1–2 risk factors and higher PRFT), intermediate-2 (1–2 risk factors but PRFT ≤ median; more than 2 risk factors but PRFT > median), poor (more than 2 risk factors and PRFT ≤ median). PRFT-modified IMDC risk model showed higher differentiating degree in predicting OS (**A**,**B**) and PFS (**C**,**D**) comparing with traditional IMDC risk model. IMDC: International Metastatic Renal-Cell Carcinoma Database Consortium. PRFT: perirenal fat thickness; OS: overall survival; PFS: progression-free survival; TDT: time from diagnosis to treatment.

**Figure 4 nutrients-14-03388-f004:**
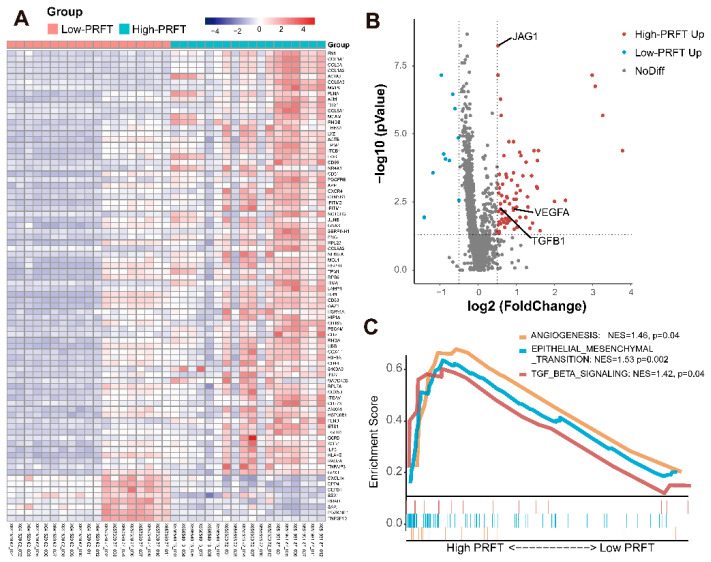
**Gene sequencing from six patients with different PRFT.** Transcriptome sequencing was performed on the RCC of six patients with different PRFT. Heat maps (**A**) and volcano figure (**B**) of differential genes was showed. GSEA analysis identified that the RCC of patients with high PRFT have increased angiogenesis gene signatures (**C**). GSEA: Gene Set Enrichment Analysis; PRFT: perirenal fat thickness.

**Figure 5 nutrients-14-03388-f005:**
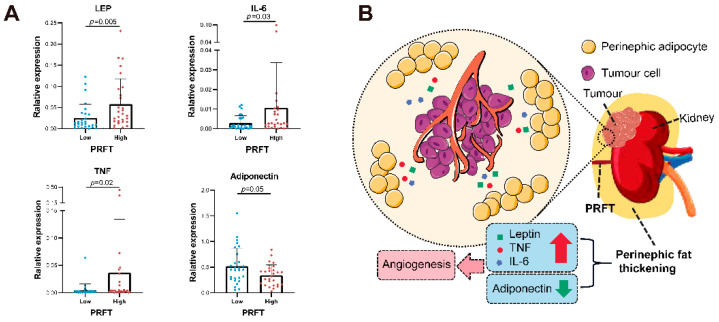
**Potential mechanisms of perirenal fat in predicting patient clinical outcomes.** Perirenal fat was collected from fifty-six patients for qPCR. We found the genes of leptin, IL-6 and TNF were highly expressed in patients with high PRFT, while low expression of adiponectin gene was showed (**A**). The possible patterns were mapped based on relevant literature and results (**B**). IL-6: interleukin-6; PRFT: perirenal fat thickness; qPCR: quantitative polymerase chain reaction; RCC: renal-cell-carcinoma; TNF: tumor necrosis factor.

**Table 1 nutrients-14-03388-t001:** Baseline characteristics.

Characteristics	All Patients (*n* = 358)
Age, median (IQR)	56 (48–64)
Male, *n* (%)	267 (74.6)
Karnofsky score < 80 *****, *n* (%)	81 (22.6)
TDT < 1 year, *n* (%)	241 (67.3)
Clear cell carcinoma, *n* (%)	249 (69.6)
Nephrectomy, *n* (%)	289 (80.7)
Immunotherapy, *n* (%)	146 (40.8)
Metastatic sites, *n*(%)	
Lung	170 (47.5)
Bone	102 (28.5)
Liver	37 (10.3)
Adrenal gland	37 (10.3)
Lymph node	168 (46.9)
Laboratory marker, median (IQR)	
Hemoglobin, g/L	128.0 (112.3–142.0)
Neutrophil, 10^9^/L	4.6 (3.5–6.8)
Platelets, 10^9^/L	245.5 (186.3–318.8)
Albumin, g/L	40.9 (35.2–44.1)
Serum calcium, mmol/L	128.0 (112.3–142.0)
First-line treatment, *n* (%)	
Sunitinib	183 (51.1)
Axitinib	68 (19.0)
Pazopanib	60 (16.8)
Sorafenib	30 (8.4)
Others	17 (4.7)
Body composition, median (IQR)	
BMI, kg/m²	23.0 (21.0–24.9)
PRFT, cm	1.6 (1.1–2.6)
Lateral	1.0 (0.6–1.5)
Posterior	0.6 (0.4–1.0)
SM, cm²	128.9 (108.6–146.9)
VAT, cm²	81.9 (39.3–138.4)
SAT, cm²	100.2 (66.1–145.4)
TAT, cm²	195.1 (112.1–290.8)
SMI, cm²	46.0 (40.4–51.5)
VAT/TAT	0.4 (0.3–0.5)

* Karnofsky score < 80 means Cancer patients cannot maintain a normal life and work. IMDC: International Metastatic Renal-Cell Carcinoma Database Consortium. IQR: interquartile range; PRFT: Perirenal fat thickness; SAT: subcutaneous adipose tissue; SM: skeletal muscle; SMI: skeletal muscle index; TAT: total adipose tissue; TDT: time from diagnosis to treatment; VAT: visceral adipose tissue.

**Table 2 nutrients-14-03388-t002:** Univariate analysis of baseline characteristics predictive for overall survival and progression-free survival.

Characteristics	Overall Survival *		Progression-Free Survival *	
	HR (95%CI)	*p*-Value	HR (95%CI)	*p*-Value
Baseline characteristics				
Age > 60	1.03 (0.72–1.49)	0.87	0.93 (0.73–1.18)	0.54
Female	1.33 (0.87–2.03)	0.19	1.13 (0.86–1.49)	0.38
Karnofsky score < 80 ******	2.86 (1.96–4.18)	<0.001	1.92 (1.47–2.52)	<0.001
Clear cell carcinoma	0.93 (0.62–1.40)	0.731	0.73 (0.56–0.94)	0.01
Treatment experience				
TDT < 1 year	2.46 (1.56–3.87)	<0.001	1.30 (1.00–1.69)	0.048
Nephrectomy	0.66 (0.4–1.06)	0.09	0.57 (0.42–0.77)	<0.001
Immunotherapy	0.47 (0.32–0.70)	<0.001	0.93 (0.73–1.19)	0.56
Laboratory marker				
Albumin < LLN, g/L	1.45 (1.01–2.09)	0.05	1.23 (0.96–1.56)	0.10
Hemoglobin < LLN, g/L	1.82 (1.25–2.66)	0.002	1.58 (1.23–2.04)	<0.001
Neutrophil > ULN, 10^9^/L	1.28 (0.87–1.88)	0.21	1.11 (0.85–1.44)	0.43
Platelets > ULN, 10^9^/L	1.38 (0.93–2.04)	0.11	1.27 (0.98–1.65)	0.08
Corrected calcium > ULN, mmol/L	4.01 (2.39–6.74)	<0.001	2.41 (1.58–3.69)	<0.001
Body composition ***				
BMI > 24, kg/m^2^ ****	0.87 (0.59–1.27)	0.46	0.89 (0.69–1.15)	0.38
PRFT > Median (1.6 cm)	0.53 (0.37–0.77)	0.001	0.75 (0.59–0.95)	0.02
SM > Median (128.9 cm^2^)	0.70 (0.49–1.01)	0.06	0.72 (0.57–0.92)	0.009
VAT > Median (81.9 cm^2^)	0.60 (0.42–0.87)	0.007	0.74 (0.58–0.94)	0.01
SAT > Median (100.2 cm^2^)	0.47 (0.33–0.69)	<0.001	0.69 (0.54–0.88)	0.002
SMI > Median (46.0 cm^2^)	0.88 (0.61–1.26)	0.49	0.85 (0.67–1.08)	0.19
TAT > Median (195.1 cm^2^)	0.56 (0.39–0.82)	0.002	0.70 (0.55–0.89)	0.004
VAT/TAT > Median (0.4)	0.97 (0.67–1.39)	0.85	0.89 (0.7–1.13)	0.33

* 117 patients had died, and 267 patients had tumor progression. ** Karnofsky score < 80 means Cancer patients cannot maintain a normal life and work. *** Dichotomies of continuous variables was sex-specific in body composition. **** BMI > 24 was overweight standards for Chinese population. BMI: body mass index; 95%CI: 95% confidence interval; HR: hazard ratio; LLN: lower limits of normal; PRFT: perirenal fat thickness; SAT: subcutaneous adipose tissue; SM: skeletal muscle; SMI: skeletal muscle index; TAT: total adipose tissue; TDT: time from diagnosis to treatment; ULN: upper limits of normal; VAT: visceral adipose tissue.

## Data Availability

The data presented in this study are available on request from the corresponding author.

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
