# Peer review of "Perirenal Fat Thickness Significantly Associated with Prognosis of Metastatic Renal Cell Cancer Patients Receiving Anti-VEGF Therapy"

_nutrients, 2022, doi:10.3390/nu14163388_

Round 1

Reviewer 1 Report

Ning et al, have examined the association between some adiposity-related variables eith the prognosis of metastatic renal cell cancer in patients receiving anti-VEGF theraphy. The paper has several methodological limitations that must be improved. This is a retrospective analysis, this being a limitation that is commented in the discussion. However more comments on this should be included in the introduction and in the abstract. Likewise, the aims of the paper must be clearly detailed in the Introduction section. Several limitations have been detected in methods. The authors must indicate the variables used in the multivariable methods to adjust for. In addition, the rational fo only including 6 patients in the gene-sequencing analysis must be explained. Sample size is very small and no power calculations or corrections for multiple genetic sites have been mentioned. Likewise, in the abstract, the authors must include the sample size for genetic analysis (n=6). In addition, the gene expression analysis and the included sample size must be better explained. Sample size here is 56 patients. Are these a random sample? Any selection criteria? Sample size calculations? Morevover in results, the section 2.1 "Gene sequencing analysis in different PRFT" is confusing because both gene sequencing results from 6 patients and gene expression analysis of selected gene by RT-PCR have been reported together. Regarding BMI, overweight has been defined and BMI>=25 kg/m2 in discussion, but in results, BMI has been analyzed as <24 or >=24. This should be explained. The revised version of the manuscript must be improved and all the Sections from the abstract to Conclusions should be updated.

Author Response

Thank you for your review comments.  We have revised the article, and we think the revised article is more suitable for Nutrients. 

Author Response

Reviewer 1: 1. This is a retrospective analysis, this being a limitation that is commented in the discussion. However more comments on this should be included in the introduction and in the abstract. Response: Thanks for your reminding. We noted at the end of the introduction and abstract that this was a retrospective study and pointed out some limitations. We now state in the introduction: “Although the retrospective design has limitations, this study suggests body composition, especially PRFT, significantly associated with prognosis of Chinese mRCC patients receiving anti-VEGF therapy.” In the abstract: “Main limitation is retrospective design.” 2. Likewise, the aims of the paper must be clearly detailed in the Introduction section. Response: Thanks for your reminding. We added the aims of the research at the end of the introduction section. “In this study, a multicenter retrospective cohort was conducted to analyze the impact of BMI and body composition on the OS and PFS in Chinese mRCC patients receiving anti-VEGF therapies. The prognostic value of fat in different areas of the body was evaluated and compared. The role of body composition played in the IMDC model was also identified in this study.” 3. Several limitations have been detected in methods. The authors must indicate the variables used in the multivariable methods to adjust for. Response: Thanks for your reminding. We added this information at the end of the method section (2.5 Statistical analysis): “In multivariable analysis, all the parameters related to CT-based body composition were adjusted for gender and IMDC score.” 4. In addition, the rational for only including 6 patients in the gene-sequencing analysis must be explained. Sample size is very small and no power calculations or corrections for multiple genetic sites have been mentioned. Response: Thank you very much for your comments. In order to provide a reasonable hypothesis for the results of clinical analysis and provide ideas for the study of related mechanisms, we collected tumor samples from six patients with renal cell carcinoma. Due to our negligence, we did not submit the results of the differential gene analysis. We are very sorry for this mistake, and we have submitted this data (Table S3). Due to the limitation of research funds, the sample size was not particularly large. However, we will use this as a basis for further research to find more evidence to support this hypothesis. We have added this as a limitation of this study in the part of discussion: “Third, although we hope to put forward reasonable hypotheses of the clinical findings, the sequencing sample size (n=6) was small.” 5. Likewise, in the abstract, the authors must include the sample size for genetic analysis (n=6). Response: Thanks for your suggestion. In the abstract, we added the sample size for genetic analysis. “Gene sequencing analysis (n=6) revealed that patients with high PRFT had increased angiogenesis gene signatures (NES=1.46, p=0.04) which might explain why better drug response to anti-VEGF therapy in mRCC patients with high PRFT.” 6. In addition, the gene expression analysis and the included sample size must be better explained. Sample size here is 56 patients. Are these a random sample? Any selection criteria? Sample size calculations? Response: Thank you very much for your comments. The sample collecting for RT-PCR from patients were random, and we have collected as many samples as possible to improve statistical power. Our selection criteria were patients with renal cell carcinoma who underwent radical nephrectomy at our hospital between July and August 2021, and the quality of extracted RNA was qualified. We have added this in the method (2.4. Quantitative reverse transcription polymerase chain reaction(RT-qPCR)): “We randomly collected perirenal fat samples from all patients with RCC who underwent radical nephrectomy at our hospital between July and August 2021(n=61). Only RNA of qualified quality was included in the analysis(n=56).” 7. Moreover, in results, the section 2.1 "Gene sequencing analysis in different PRFT" is confusing because both gene sequencing results from 6 patients and gene expression analysis of selected gene by RT-PCR have been reported together. Response: Thanks for your reminding. In order to develop possible hypotheses for clinical outcomes, gene sequencing from 6 patients and RT-PCR of selected gene from 56 patients were performed in our study. To prevent misunderstanding, we have divided the two parts into two separate figures. 8. Regarding BMI, overweight has been defined and BMI>=25 kg/m2 in discussion, but in results, BMI has been analyzed as <24 or >=24. This should be explained. Response: Thank you very much for your comments. In European countries and USA, overweight has been defined and BMI≥25 kg/m2 . Many previous studies of BMI and metastatic renal cell carcinoma have been based on European and American populations. However, according to Chinese standard, a BMI greater than 24 is considered overweight[1], and this is a study based on Chinese metastatic renal cell carcinoma. We add remarks in table2: “a BMI greater than 24 is considered overweight according to Chinese standard."

Reviewer 2 Report

Ning et al conducted a retrospective analysis of 358 patients with metastatic renal cell cancer (mRCC) receiving anti-VEGF therapy in 3 hospitals in China. Specifically the association of various measures of adiposity (BMI, visceral adipose tissue, subcutaneous adipose tissue, perirenal fat thickness) with mortality and disease free survival were explored and compared/cotrasted. The added value of including these adiposity measures on top of the International Metastatic Renal Cell Carcinoma Database Consortium (IMDC) model was also explored. Finally, the authors performed transcriptome sequencing on the RCC of subset of the patients as well as gene sequencing  of the perirenal adipose tissue of another subset of patients. 

The  authors should be commended for research to potentially improve outcomes for mRCC patients. Their efforts to understand the biologic mechanisms behind their clinical findings was also of interest.

Several major comments/questions were noted (other minor comments also included in the attached annotated pdf):

1. The background lacked a clear and compelling rationale for why the study was undertaken. In particular, I felt more text was needed setting up the relationship between BMI and other adiposity measures as well as the potential for misclassification among the Asian population. There is also quite a bit of text in the  discussion that better fits the background

2. Related to the above, the study does not include a clear/testable hypothesis (or objectives)

3. There are a number of improvements that should be considered for the analytic methods:

a. The correlation of the various adiposity measures needs to be included

b. The functional form of the adiposity measures should be explored. The current parameterization (binary cutpoint at median may not be be appropriate, and may completely explain the findings; this also makes it very difficult for subsequent studies to compare)

c. The language used to describe statistical findings in many cases should be more precise (e.g. "improved model fit as measured by AIC" rather than "better predictive abilities")

4. Median survival should be reported

Reviewer 2:
1. The background lacked a clear and compelling rationale for why the study was undertaken. In particular, I felt more text was needed setting up the relationship between BMI and other adiposity measures as well as the potential for misclassification among the Asian population. There is also quite a bit of text in the discussion that better fits the background
Response: Thank you very much for your comments. Actually, BMI has been reported to be a significant predictor for better prognosis in patients with metastatic renal cell cancer (mRCC). However, those studies were almost conducted among Caucasian who tend to have a higher BMI than Asian. In Asian patients who tend to have lower BMI, the performance of BMI to identify excess adiposity has a high specificity and low sensitivity, suggesting an underdetection of obesity. In our analysis, we also found BMI was not significant with prognosis in mRCC patients receiving antiVEGF therapy. However, computed tomography (CT)-based body composition measures can distinguish between visceral adipose tissue (VAT), subcutaneous adipose tissue (SAT), and perirenal fat thickness (PRFT). CT-based body composition measure was a better way to evaluate fat distribution and nutrient levels. Besides, fat in different areas of the body also has different biological properties. Hence, it worth exploring whether fat in different areas of the body is a more valuable predictor in mRCC comparing with BMI.
We have revised and improved the part of introduction in our article. We believe that the revised introduction clearly illustrated the purpose of our research.
“The body mass index(BMI), a traditional indicator of general obesity, was reported to be a significant predictor for overall survival(OS) and progression-free survival(PFS) in mRCC patients receiving anti-VEGF therapies[7,8]. However, most studies between BMI and mRCC have been conducted among Caucasian, who tend to have a higher BMI than Asian. In Asian patients who tend to have lower BMI, the performance of BMI to identify excess adiposity has a high specificity and low sensitivity, suggesting an underdetection of obesity[9].Besides, there was a high 
variability in fat mass and muscle mass within all strata of BMI in cancer patients[10], and BMI failed to reflect more specific measures of body composition[11]. For example, a Japanese study have reported that Asian patients with lower BMI had better prognosis than those who were overweight in gastrointestinal cancer surgery[12], and this finding were quite different from previous American studies of gastrointestinal cancer among Caucasian [13].
Computed tomography(CT)-based body composition measures can distinguish between visceral adipose tissue(VAT), subcutaneous adipose tissue(SAT), and perirenal fat thickness(PRFT)[14]. Since BMI has been shown to be an imprecise measurement of fat-free and fat mass, body cell mass and fluids, CT-based body composition measure was a better way to evaluate fat distribution and nutrient levels[15]. Besides, fat in different areas of the body also has different biological properties. VAT and SAT have different origins and the developmental heterogeneity may determine the functional heterogeneity[16]. Compared with SAT, VAT has more blood vessels, innervation, inflammatory cells, is metabolically more active, more sensitive to lipolysis and more insulin-resistant[17]. Although perirenal fat is a type of VAT adjacent to the kidneys, it is reported to originate from both adipocyte precursor cells and mature adipocytes; different from the origins of other components of VAT[18,19]. Chen et al. found that the anatomical position and developmental heterogeneity of perirenal fat make it more sensitive to the development of chronic kidney disease than VAT and SAT in patients with diabetes[20]. However, the association between fat in different areas of the body and anti-VEGF therapy in patients with mRCC remains unclear. Given the superiority of CT-based body composition and special biological properties of fat, it worth exploring whether fat in different areas of the body is a more valuable predictor in mRCC comparing with BMI.”
2. Related to the above, the study does not include a clear/testable hypothesis (or objectives)
Response: Thanks for your reminding. We added the aims of the research in the end of the introduction section:
“In this study, a multicenter retrospective cohort was conducted to analyze the impact of BMI and body composition on the OS and PFS in Chinese mRCC patients receiving anti-VEGF therapies. The prognostic value of fat in different areas of the body was evaluated and compared. The role of body composition played in the IMDC model was also identified in this study.”
3. There are a number of improvements that should be considered for the analytic methods:
a. The correlation of the various adiposity measures needs to be included
Response: Thanks for your reminding. We added this analysis, as shown in the figure below.
“ Figure S2. The correlation of the various adiposity measures. There are correlations between different adiposity measures. However, the correlation between BMI and PRFT was less than 50%, while there is a high correlation (72%) between VAT and PRFT.”
b. The functional form of the adiposity measures should be explored. The current parameterization (binary cutpoint at median may not be be appropriate, and may not completely explain the findings; this also makes it very difficult for subsequent studies to compare)’? 
Response: Thanks for your reminding. The dichotomization of parameters is conducive to the classification of patients into high and low risk groups, which is convenient for clinical application. Binary cutpoint at median is a common 
dichotomous method for data analysis[2-4]. In our analysis, we looked for the best cutoff to predict survival, but found it to be close to the median. Hence, we choose binary cutpoint at median to make it easier to explain. 
c. The language used to describe statistical findings in many cases should be more precise (e.g. "improve model fit as measured by AIC" rather than "better predictive abilities")
Response: I’m very sorry for making such a mistake. We have further refined the English of the article in order to improve its readability.In the method (2.5. Statistical analysis): “The improved model fit was measured by the concordance index(C-index), area under curve (AUC), and Akaike Information Criterion (AIC). Conclusively, the higher C-index and AUC or the lower the AIC, the better the model fit.”In the result (3.3. Multivariable Analysis): “The C-index for the IMDC risk model significantly demonstrated OS (0.69, 95% confidence interval, 95% CI: 0.63-0.74) and PFS (0.61, 95%CI: 0.57-0.65). BMI-modified IMDC model didn’t significantly improve the prediction efficiency of OS (71.96%) and PFS (70.45%).”In the result (3.4. PRFT-modified IMDC model): “PRFT modified IMDC model showed a higher degree of fit comparing with traditional IMDC model in predicting OS (Figure 3A&B, AIC: 1167.32 vs 1181.53) and PFS (Figure 3C&D, 2640.79 vs 2657.92).”
4. Median survival should be reported
Response: Thanks for your reminding. In the first paragraph of result, we 
described the overall median survival time (3.1. Patient Characteristics and Outcomes): “The median OS and PFS for the entire cohort of 358 patients were 22.0(IQR: 13.0-37.0) months and 9.1(IQR: 4.8-16.1) months.” Besides, median survival has been added in each figure in this article.

Besides, we have responded to the other question notes in the PDF and made modifications in the manuscript.
1. Appropriate body-mass index for Asian populations and its implications for policy and intervention strategies. Lancet 2004, 363, 157-163, doi:10.1016/s0140-6736(03)15268-3.
2. Shimonov, M.; Abtomonova, Z.; Groutz, A.; Amir, H.; Khanimov, I.; Leibovitz, E. Associations between body composition and prognosis of patients admitted because of acute pancreatitis: a retrospective study. Eur J Clin Nutr 2021, 75, 817-822, doi:10.1038/s41430-020-00789-y.
3. Gianni, M.L.; Bettinelli, M.E.; Manfra, P.; Sorrentino, G.; Bezze, E.; Plevani, L.; Cavallaro, G.; Raffaeli, G.; Crippa, B.L.; Colombo, L., et al. Breastfeeding Difficulties and Risk for Early Breastfeeding Cessation. Nutrients 2019, 11, doi:10.3390/nu11102266.
4. Alvarez Secord, A.; Bell Burdett, K.; Owzar, K.; Tritchler, D.; Sibley, A.B.; Liu, Y.; Starr, M.D.; Brady, J.C.; Lankes, H.A.; Hurwitz, H.I., et al. Predictive Blood-Based Biomarkers in Patients with Epithelial Ovarian Cancer Treated with Carboplatin and Paclitaxel with or without Bevacizumab: Results from GOG-0218. Clin Cancer Res 2020, 26, 1288-1296, doi:10.1158/1078-0432.Ccr-19-0226

Round 2

Reviewer 1 Report

The manuscript has been revised and corrected

Reviewer 2 Report

The authors should be commended for extensive editing based on reviewer comments. However, I find the manuscript still lacks a clear hypothesis and the main conclusions (PRFT improves prediction OS and PFS compared to traditional adiposity measures) may simply be due to the parameterization of PFT vs. other adiposity measures.